# Contrastive Patient-level Pretraining Enables Longitudinal and Multimodal Fusion for Lung Cancer Risk Prediction

**Thomas Z. Li**[1,2]  (ID)     THOMAS.Z.LI@VANDERBILT.EDU
[1] *Department of Biomedical Engineering, Vanderbilt University, Nashville, TN*
[2] *Medical Scientist Training Program, Vanderbilt University, Nashville, TN*
**Lianrui Zuo**[3]  (ID)     LIANRUI.ZUO@VANDERBILT.EDU
[3] *Department of Electrical and Computer Engineering, Vanderbilt University, Nashville, TN*
**Yihao Liu**[3]     YIHAO.LIU@VANDERBILT.EDU
**Aravind R. Krishnan**[3]     ARAVIND.R.KRISHNAN@VANDERBILT.EDU
**Kim L. Sandler**[4]     KIM.SANDLER@VUMC.ORG
[4] *Department of Radiology, Vanderbilt University Medical Center, Nashville, TN*
**Thomas A. Lasko**[3,5]  (ID)     TOM.LASKO@VANDERBILT.EDU
[5] *Department of Biomedical Informatics, Vanderbilt University, Nashville, TN*
**Fabien Maldonado**[6]     FABIEN.MALDONADO@VUMC.ORG
[6] *Department of Medicine, Vanderbilt University Medical Center, Nashville, TN*
**Bennett A. Landman**[1,3,4,5]     BENNETT.LANDMAN@VANDERBILT.EDU

**Editors:** Accepted for publication at MIDL 2025

## Abstract

Leveraging longitudinal and multimodal data is important for clinical predictive tasks. Contrastive language-image pretraining (CLIP) has been successful in learning multimodal representations by aligning paired images and captions, i.e. medical images and corresponding radiology report. However, in real clinical settings, the alignment of unpaired modalities, such as medical images and clinical notes collected at different times, is an open challenge, even though such data are ubiquitous in practice. This study conducts contrastive pretraining between longitudinal chest CTs and clinical variables on the patient level using a large public lung cancer screening dataset. Leveraging a time-distanced transformer to encode longitudinal imaging and an open-source text embedding to encode clinical variables, we optimize contrastive loss between the embedded modalities from same patient (positive pair) against those from different patients (negative pair). We find that finetuning the CLIP representation significantly improves prediction of lung cancer risk in two types of clinical populations (0.895 and 0.893 AUC) compared to conventional multimodal fusion (0.873 and 0.875 AUC) and single modality baselines. These results demonstrate how contrastive patient-level pretraining can enable longitudinal and multimodal fusion without additional training data. We released our code and pre-trained weights at https://github.com/MASILab/lung-cplp.

**Keywords:** contrastive language-image pretraining (CILP), multimodal, chest CT, lung cancer

## 1. Introduction

Leveraging information across time and modalities is crucial for many clinical problems that machine learning can have a significant impact. However, multimodal models often struggle with overfitting, and are susceptible to both intra- and inter-modality variation compared to simpler imaging models (Wang et al., 2020; Tejero-de Pablos, 2024). Furthermore, modalities collected from routine clinical practice are irregularly sampled and asynchronous with one another. Because of these challenges, researchers often find that longitudinal multimodal models are prohibitively data-hungry (Udandarao et al., 2024) and frequently fail to generalize to different clinical settings (Li et al., 2024; Lasko et al., 2024b). Unsupervised and self-supervised learning offer promising solutions by leveraging large-scale data to enhance multimodal representation and improve model robustness (Li et al., 2023a).

Contrastive pretraining is a prominent self-supervised paradigm that leverages a contrastive loss to encourage one embedding to be similar to a related embedding while being less similar than an arbitrarily selected pair (van den Oord et al., 2019; Jia et al., 2021). In contrastive language-image pretraining (CLIP) (Radford et al., 2021; Zhang et al., 2022), the related pairs consist of an image and its natural language description, such as a CT study and radiology report, but the pairs can be defined using any useful relationship. Lee et al. (2022) included intra-modality pairs by aligning pairs of augmentations on the same image and Lin et al. (2023) included figure-caption pairs from web-scale biomedical documents.

Applications of CLIP on medical data are concentrated in radiology and pathology where image-text pairs are abundant, resulting in promising downstream performance (Huang et al., 2021; Chen et al., 2024; Liu et al., 2023; Jang et al., 2024; Lu et al., 2024). Many challenging clinical problems however, such as prediction of future lung cancer risk, rely on multiple modalities not because they describe each other well but because *they do not* (Acosta et al., 2022). Solving such tasks, to some degree, may rely on integrating probabilistically independent sources that are exclusively found in different modalities (Lasko et al., 2024a). For the lung cancer domain, family history of lung cancer is one of many well-known risk factors that does not have a direct CT correlate. Beyond radiologist- and pathologist-annotated reports, most of the clinical record does not directly describe medical images. This begs the largely unexplored question as to whether unpaired modalities can be aligned simply based on being derived from the same patient and same relative time.

In this study we leverage a time-distance chest CT encoder and an open-sourced text representation model to compute embeddings for consecutive screening chest CTs and clinical variables respectively for each patient. We conduct contrastive pretraining across the image and text embeddings by defining positive and negative pairs on the patient ID. This encourages embeddings of the unpaired modalities to be similar, which admittedly is counterintuitive. Nevertheless we hypothesized that the representation learned from CLIP would achieve better separation in downstream classification than supervised multimodal representations. We show that contrastive patient-level pretraining leads to improved lung cancer risk prediction after finetuning compared to supervised training on the same training set using the same model architecture.

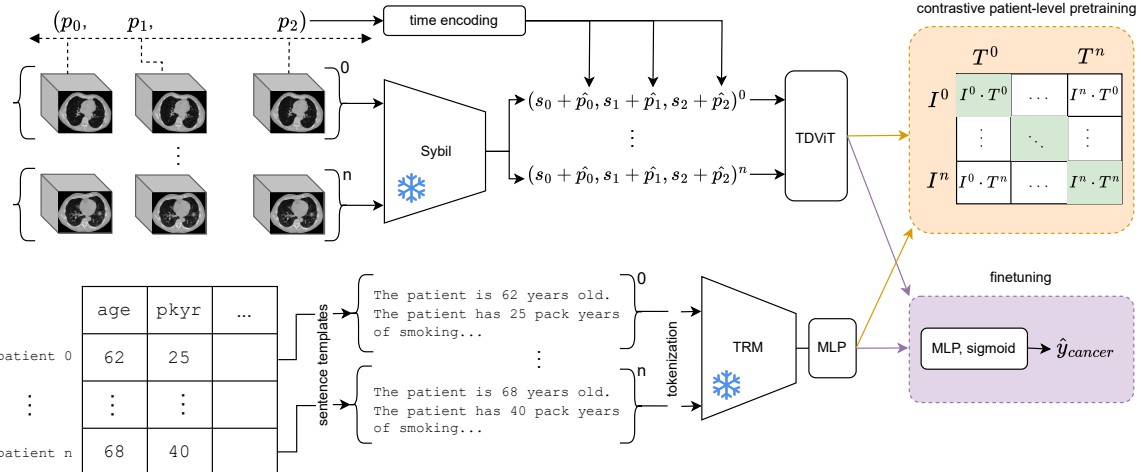

Figure 1: Our approach ingests up to three annual chest CTs and cross-sectional clinical variables. Top: The chest CTs collected at times $(p_0, p_1, p_2)$ are transformed into feature vectors $(s_0, s_1, s_2)$ using a pretrained ResNet (Sybil). Each feature vector is added to its corresponding time encoding $\hat{p}_i$ to form a sequence for input into a time-distance transformer (TDViT). The embedding corresponding to the latest time point was used as the image embedding, $I^i$, for CLIP and finetuning. Bottom: Tabular variables are transformed into a natural language form using sentence templates. We used a pretrained text representation model (TRM) and a multi-layer perceptron (MLP) to compute the text embedding, $T^i$, for CLIP and finetuning. During both stages, weights of Sybil and TRM were frozen.

## 2. Approach

### 2.1. Patient-level Embeddings

This study focused on learning patient-level embeddings of public chest CTs and clinical variables from the National Lung Screening Trial (NLST) (NLST, 2011). We reproduced the test set from (Ardila et al., 2019) and used all other subjects for training. The clinical variables are collected through surveys and chart reviews at the time of trial randomization. Following randomization, patients annually receive up to three low-dose chest CTs (Table 1). Unlike the paired modalities conventionally seen in CLIP studies, the clinical variables are chronologically out of sync with the collected imaging in the NLST for up to four years in some cases. Moreover, they are semantically unpaired, meaning that the clinical variables do not directly describe the CT image. Following CLIP, we leverage modality-specific encoders (Figure 1).

#### 2.1.1. IMAGE ENCODER

The image encoder is composed to two modules. First, CT slices were normalized to a Hounsfield window of $[-600, 1500]$ and resized to $512 \times 512$. These were given to Sybil, a ResNet developed to detect lung cancer on lung screening chest CTs (Mikhael et al., 2023),

Table 1: Training and Testing Sets from NLST. `COPD`: Presence of chronic obstructive pulmonary disease; `phist`: Personal history of cancer, `fhist`: Family history of lung cancer.

| | # Patients (# Cases) | # Chest CTs | Clinical Variables |
|---|---|---|---|
| Training[†] | $22,571$ (598) | $61,631$ | age, sex, race, BMI*, |
| Screening Test | $2,315$ (94) | $6,615$ | smoking quit time, smoking |
| Nodule Test | $2,057$ (93) | $5,879$ | duration, smoking pack |
| | | | years, COPD*, emphysema, |
| | | | phist., fhist., smoking status |

[†]Used in both pretraining and finetuning stages. *Variable missing in $< 1\%$ of subjects

which resulted in $\mathbb{R}^{256}$ feature vectors for each chest CT. Second, we follow the strategy of time-distance transformers (Li et al., 2023c,b) which added time embeddings representing the relative days from the earliest scan to the corresponding feature vectors. This allows the encoder to learn longitudinally, such as observing the change in lung nodules features over time. We set the sequence length to 3 and we used attention masks to avoid attending to missing tokens in the sequence (i.e. when the patient received less than three chest CTs). During training, the ResNet weights were frozen while time-distanced transformer weights were optimized (Figure 1, top).

### 2.1.2. TEXT ENCODER

To encode the clinical variables, we leveraged a pretrained BERT-style (Zhang et al., 2024b) text representation model (TRM) from Hugging Face, `dunzhang/stella_en_400M_v5`, selected for its small memory footprint and strong performance on the Massive Text Embedding Benchmark (Muennighoff et al., 2023). The text encoder's input consisted of the patient's tabular variables transformed into a natural language paragraph using sentence templates (Figure 1, bottom). Units of measurement were included with numerical variables and missing variables were simply left out of the paragraph. In our experiments, the natural language transform outperformed transformation into JSON format, but it was unclear if this was specific to the TRM we selected. The set of clinical variables included the length of lung cancer-free follow up and if the patient ultimately developed lung cancer. The TRM tokenized the natural language paragraph (Conneau et al., 2020; Kudo and Richardson, 2018) and computed text embeddings. Lastly, a multi-layer perceptron (MLP) processed the resulting text embeddings. Although this study only accessed cross-sectional clinical variables, the MLP can be replaced with a transformer in future studies where the non-imaging data is longitudinal. During training, the weights of the TRM were frozen while the MLP weights were optimized (Figure 1, bottom).

### 2.2. Contrastive Patient-level Pretraining

We hypothesized that representations learned through contrastive patient-level pretraining would be more useful in lung cancer classification compared to representations learned through supervised learning. In order to align imaging and clinical variables using con-

trastive pretraining, we arranged image and text embedding pairs for each patient. The image embedding was the token corresponding to the chronologically latest chest CT while the text embedding was simply the output of the text encoder. Within a batch, an image and text embedding was considered a positive pair if they corresponded to the same patient or negative if the image and text embeddings were derived from different patients. All pairs were multi-modal (i.e. we did not allow pairs to be of image-to-image or text-to-text embeddings). We performed CLIP to maximize to cosine similarity between positive pairs via symmetric cross entropy loss. We employed Adam optimization (Kingma and Ba, 2017) and a learning rate with cosine annealing and warm restart cycles (Loshchilov and Hutter, 2017). Training was stopped when no improvement in the loss was observed after 1000 epochs. This stage of training was highly sensitive to batch size. On one hand, a large batch size increased the number of contrasted pairs and reduced variation within the batch. However, batch sizes beyond a certain limit underperformed, perhaps due to the model taking very few steps per epoch (Zhang et al., 2024a). We found that the critical batch size for our training dataset was 2401 which corresponded to 10 steps per epoch. We froze aforementioned parts of the image and text encoder and trained with mixed precision (Micikevicius et al., 2018) to accommodate our desired batch size. Code and pre-trained models are released at https://github.com/MASILab/lung-cplp.

## 3. Experiments and Results

We examined the CLIP embeddings by testing its zero-shot and fine-tuned prediction of 2-year lung cancer risk on a withheld test set. We compared against baselines to examine the benefit of contrastive patient-level pretraining. Each experiment was performed in two clinical cohorts: the entire test set (screening) and the subset of patients with a lung nodule detected (detected-nodules). In brief, the screening cohort tests the model's ability to estimate future lung cancer risk, regardless of whether a nodule is present, while the detected-nodule cohort evaluates the model's ability to discriminate between benign and malignant nodules. We report the mean AUC and 95% confidence interval from 1000 bootstrapped samples of the predictions, sampling with replacement from the test set. A two-sided Wilcoxon signed-rank test was used to test if the mean AUCs of each approach was significantly different than the rest for $p < 0.05$.

All experiments were conducted in Python 3.12 using Pytorch 2.5 and CUDA 12.4 on a Ubuntu 22.04 server with a single NVIDIA RTX 6000.

### 3.1. Zero-shot Classification

We were interested in evaluating how useful the pretrained patient-level embeddings were for predicting 2-year lung cancer risk out-of-the-box. In contrast to other zero-shot tasks, the concept of lung cancer is not entirely unseen during pretraining since the lung cancer label and follow up period was included during CLIP training. However, the task of predicting two-year lung cancer risk still presents a challenging domain shift both in terms of longitudinal semantics and class prevalence in the test set, as demonstrated by the zero-shot results to be presented. To evaluate the performance of zero-shot classification on the test set, we converted each diagnostic label into a templated sentence, "The patient [developed/did not develop] lung cancer after 2 years". This was appended to the patient's

Table 2: Prediction of 2-year lung cancer risk on test cohort (mean AUC [95% CI]). `cross-sec.`: cross-sectional, `long.`: longitudinal.

| | Screening ($n = 2,315$) | Detected-Nodule ($n = 2,057$) | Imaging type | Non-imaging type |
|---|---|---|---|---|
| Sybil-cs | 0.877 [0.872, 0.881] | 0.869 [0.864, 0.874] | cross-sec. | N/A |
| Sybil-TDViT | 0.861 [0.860, 0.863] | 0.858 [0.856, 0.859] | long. | N/A |
| MM-tabular | 0.871 [0.869, 0.872] | 0.876 [0.871, 0.880] | long. | tabular |
| MM-lang | 0.873 [0.871, 0.874] | 0.875 [0.873, 0.876] | long. | language |
| CLIP-tabular | 0.849 [0.843, 0.854] | 0.837 [0.831, 0.842] | long. | tabular |
| CLIP-zeroshot | 0.540 [0.538, 0.541] | 0.523 [0.521, 0.524] | long. | language |
| **CLIP-finetune** | **0.895 [0.894, 0.896]**[*] | **0.893 [0.892, 0.894]**[*] | long. | language |

[*]$p < 0.05$ against all other methods.

natural language paragraph, preserving the other clinical variables but in place of the diagnostic label. We then computed the cosine similarity between each image embedding and both possible text embeddings. The text embedding with the higher similarity score was chosen as the model's prediction in a zero-shot manner. In this setting, the CLIP approach struggled to predict 2-year lung cancer risk with performances of 0.540 and 0.523 in screening and detected-nodule settings respectively. (Table 2, CLIP-zeroshot)

### 3.2. Finetune Classification

We asked how the pretrained patient-level embeddings would perform in a supervised classification setting. Because lung cancer labels were available for each patient, the fine-tune phase used the same training set as the pretraining phase. We removed sentences pertaining to lung cancer and follow-up period, but other preprocessing steps remained unchanged. The embeddings from the image and text encoder were concatenated and fed through a MLP and sigmoid classifier. We set the learning rate of the image and text encoders to 1/10th of the MLP classifier's learning rate. This finetuned approach achieved a mean AUC of 0.895 in the lung screening setting and 0.893 in the detected-nodule setting (Table 2, CLIP-finetune).

In addition, we finetuned and evaluated four checkpoints along the course of pretraining CLIP to probe how classification performance changed with length of training (Figure 2) in the screening cohort. When evaluating at different checkpoints over 10,000 pretraining epochs, we observed that finetuning as early as epoch 2500 led to improved performance over supervised baselines (0.887, 95% CI:[0.886, 0.888] AUC). The best performance was achieved at 9000 epochs (Table 2, CLIP-finetune), albeit the finetune performance did not increase monotonically over training epochs.

### 3.3. Baselines and Ablations

Lung cancer labels are available for all patients in the NLST, which allowed us train all baselines with the same set used in the contrastive pretraining and finetuning phases. As a direct ablation of CLIP, we reproduced the architecture from the finetune stage and trained the model from random weights (Table 2, MM-lang), preserving weights that were originally

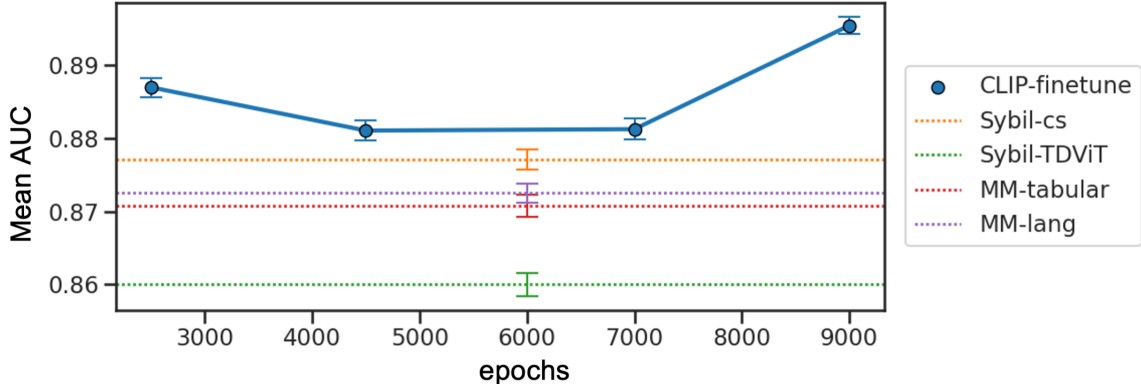

Figure 2: Four checkpoints were finetuned and evaluated over contrastive pretraining epochs. Above-baseline performance was observed at 2500 epochs and the best performance was achieved at 9000 epochs. Error bars represent 95% confidence intervals

frozen during CLIP training. Specifically we optimized the TDViT and MLP proceeding the TRM while the ResNet and TRM remained frozen. All supervised training was conducted with a batch size of 2401 and training was stopped when the validation loss failed to improve for 40 epochs. Our CLIP approach outperformed its supervised counterpart by about 0.02 AUC, with MM-lang underperforming at 0.873 AUC on the screening cohort and 0.875 AUC on the detected nodule cohort, indicating that the CLIP was responsible for the observed performance gains.

We also ablated the TRM from the text encoder, using the tabular clinical variables as input to a CLIP approach (Table 2, CLIP-tabular) and purely supervised approach (Table 2, MM-tabular). We employed multiple linear imputation (Murray, 2018) for missing variables in tabular form. We trained a longitudinal imaging baseline from random weights using the image encoder itself (Table 2, Sybil-TDViT) appended to an MLP classifier. Lastly we trained a cross-sectional imaging baseline using Sybil features from the most recent chest CT and a MLP classifier (Table 2, Sybil-cs).

MM-tabular (0.871 and 0.876 AUC) and cross-sectional imaging model Sybil-cs (0.876 and 0.869 AUC) both matched the performance of MM-lang on screening and detected-nodule cohorts respectively. The longitudinal imaging model Sybil-TDViT slightly under-performed all other methods at (0.861 and 0.858 AUC) in both clinical cohorts.

## 4. Discussion and Limitations

This work investigates contrastive pretraining with repeat chest CTs and clinical variables that are matched on patient and time but not paired semantically. The learned patient-level embeddings were evaluated based on zero-shot and finetune prediction of 2-year lung cancer risk in screening and detected-nodule cohorts. Without additional training examples, our approach, when finetuned, improved significantly over a supervised counterpart and single modality baselines.

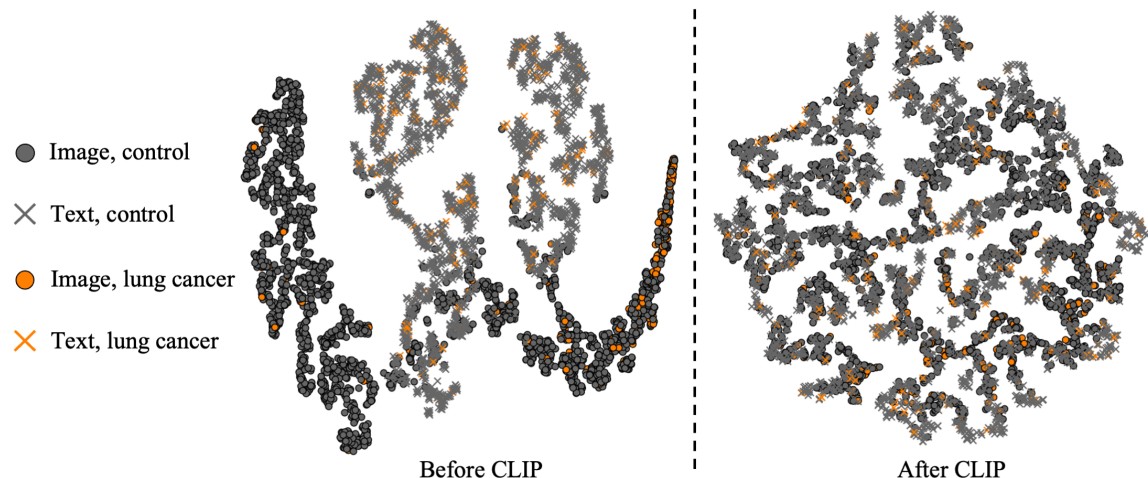

Figure 3: Image and text embeddings before (left) and after (right) CLIP were visualized with t-SNE. Without CLIP, partly-disjoint clusters can be observed within the same modality. After CLIP, image and text embeddings are pulled together. In both examples, lung cancer cases are not easily separable from controls, but the CLIP representation outperformed supervised baselines after finetuning.

To interpret the learned representation before and after CLIP, we computed image and text embeddings with Sybil and the TRM respectively using their original weights and compared them to the feature embeddings from CLIP encoders. A 2D t-SNE (van der Maaten and Hinton, 2008) map of the embeddings shows how the original embeddings cluster by modality while the image and text embeddings from CLIP are pulled together (Figure 3).

Interestingly, the supervised multimodal models, MM-lang and MM-tabular, were not able to leverage the predictive signal that we suspect is present in clinical variables to outperform the imaging classifier, Sybil-cs. Overfitting on the multimodal feature space and additional model parameters is a probable explanation for this (Wang et al., 2020; Tejero-de Pablos, 2024). Contrastive patient-level pretraining overcame these challenges to some degree, suggesting that it learned a more effective joint representation of longitudinal imaging and non-imaging modalities. Even when operating with purely labeled data, this may be a useful pretraining step for warming up downstream supervised training.

The poor zero-shot performance after CLIP warrants discussion. Given that we only included 13 variables, we did not expect our zero-shot performance to be similar to large scale foundation models. Still, these results suggest that the patient-level embeddings are not easily separable into the diagnostic classes by cosine similarity alone. Instead, a non-linear classifier, e.g. a finetuning stage, is required to fully leverage the utility of the learned representation. Figure 3 visually corroborates this as the lung cancer cases and controls do not separate into noticeable clusters. These findings could be a consequence of aligning unpaired modalities and the unseen task-learning capability of the model would be improved by covering a larger and more diverse set of clinical concepts during pretraining.

In conclusion, this paper demonstrates that contrastive pretraining enhances the fusion of longitudinal and multimodal medical data without requiring semantically paired modalities or additional training examples. To ensure applicability to real clinical data, we integrate time encodings for longitudinal information and a TRM for embedding non-imaging modalities. In data-limited settings, contrastive patient-level pretraining facilitates multimodal fusion, mitigating the overfitting challenges typically encountered with supervised approaches.

## Acknowledgments

This research was funded by the NIH through F30CA275020, 2U01CA152662, and R01CA253923-02, as well as NSF CAREER 1452485 and NSF 2040462. This study was also funded by the Vanderbilt Institute for Surgery and Engineering through T32EB021937-07, the Vanderbilt Institute for Clinical and Translational Research through UL1TR002243-06, and the Pierre Massion Directorship in Pulmonary Medicine. We utilize generative AI to generate code segments based on task descriptions, as well as to assist with debugging, editing, and autocompleting code. Additionally, generative AI has been employed to refine sentence structure and ensure grammatical accuracy. However, all conceptualization, ideation, and prompts provided to the AI stem entirely from the authors' creative and intellectual efforts. We take full responsibility for reviewing and verifying all AI-generated content in this work.

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
