# OpenReview forum: "Contrastive Patient-level Pretraining Enables Longitudinal and Multimodal Fusion for Lung Cancer Risk Prediction"
_MIDL.io/2025/Conference — MIDL 2025 Poster_

### Official Review · Reviewer_YqoU · 2025-02-19

**Confidence:** 4
**Preliminary Rating:** 3
**Final Rating:** 4

**Summary:**

This paper explores a pre-training technique for bimodal data that are not necessarily aligned, specifically longitudinal chest CT scans and clinical variables for each patient.
The core idea is to use a contrastive technique to align image and clinical representations of the same patient.
Longitudinal chest CT representations are first extracted using a pre-trained ResNet (Sibyl) and, along with their corresponding acquisition times, are fed into a Time-Distance (TDViT) Transformer.
Meanwhile, clinical data are transformed into sentence templates and processed through a pre-trained text tokenizer (TRM).
The authors then evaluate the resulting model for predicting the risk of developing lung cancer within two years—first using a zero-shot approach, as in CLIP, and subsequently by concatenating image and clinical representations as input to an MLP in a supervised learning setting.

**Strengths:**

- The paper addresses the challenge of unpaired multimodal data, a critical issue in both the medical domain and large-scale data scenarios.
- The authors enhance research transparency by using a public dataset and openly sharing their code and pre-trained model, fostering reproducibility and further advancements in the field.
-  The paper includes a comparative analysis with baselines,  although it is more an ablation, to highlight the relevance of the proposed method.
- The paper is well-structured, making it accessible and easy to follow.
- The use of masked attention to handle missing CT scans is a well-thought approach that strengthens the method’s robustness.

**Weaknesses:**

- My main concern with this work is the handling of clinical data as text. While the paper frequently references image-language data ($T^i$ for the embedding) , the study itself primarily involves longitudinal CT scans and structured clinical variables, with no explicit text data. Instead, the authors employ sentence templates to encode clinical information, which is then processed by a text encoder. Thus, the role of the TRM is to extract key information from these templated sentences—information that is already known, namely the clinical data, which are the only tokens varying across samples. This makes the approach somewhat ouroboric. CLIP’s original implementation. Benefits from the diversity and open-ended nature of text data, enabling the model to learn underlying semantic structures, which is simply not the case here.
- The evaluation methodology remains unclear in certain aspects. For instance, it is not explicitly stated how many epochs the models are fine-tuned (Section 3.2). Additionally, during this phase, are TDViT and the MLP at the TRM output (MLP-TRM) frozen (Section 3.2)? Similar clarifications are needed for the baselines and ablations (Section 3.3).
- Although using a public dataset, the authors provide little comparison with the literature. Moreover, considering that the proposed technique is a pre-training technique, we could have imagined evaluating the model on other tasks or other datasets for which results can be obtained on this dataset. For just a few examples: [2,3].
- Regarding Section 3.1 (Zero-shot Classification): as acknowledged by the authors, this is not truly zero-shot, since the cancer label and follow-up time are included in the training process. Given this setup, the reported poor performance is particularly concerning. Once these variables are available, simple discriminative rules should suffice to determine the classification label at the 2-year mark. This raises questions about the effectiveness of the pre-training process. Are the validation loss functions behaving as expected? Additionally, top-k metrics are necessary to assess whether the training is properly aligning embeddings by bringing patient representations closer together.



[1] Wang, Z., Wu, Z., Agarwal, D., & Sun, J. (2022). MedCLIP: Contrastive learning from unpaired medical images and text. EMNLP 2022

[2] Chao, H., Shan, H., Homayounieh, F., Singh, R., Doda Khera, R., Guo, H., Su, T., Wang, G., Kalra, M. K., & Yan, P. (2021). Deep learning predicts cardiovascular disease risks from lung cancer screening low dose computed tomography. Nature Communications

[3] Heuvelmans, M. A., van Ooijen, P. M. A., Ather, S., Silva, C. F., Han, D., Heussel, C. P., ... & Oudkerk, M. (2021). Lung cancer prediction by deep learning to identify benign lung nodules. Lung Cancer

**Detailed Comments:**

- The handling of the time variable in the image encoder is unclear. Is the time concatenated with the embedding obtained? Is it first projected by an MLP before being added as an encoding?
- The authors state that *"This stage of training was highly sensitive to batch size."* In the pre-training phase. Theoretically, contrastive learning benefits from larger batch sizes, as this better approximates the mutual information to be minimized (see [4]). Given this, the authors should provide their ablation study on batch size and discuss this behavior.
- The authors should provide more experience, particularly in the area of modality retrieval, or their method could also be used to find missing information in clinical data, which would be a big plus.
- The sentence *"In our experiments, the natural language transform outperformed transformation into JSON format, but it was unclear if this was specific to the TRM we selected."* needs to be rephrased or explained.
- The authors use the formulation *"finetuning”*, whereas the literature tends to use *“fine-tuning”* instead.
- Some minor English errors.

[4] Oord, A. v. d., Li, Y., & Vinyals, O. (2018). Representation learning with contrastive predictive coding.

**Justification Of The Final Rating:**

As mentioned in my comment, the authors have answered the questions raised in the revised manuscript and in their rebuttal.
- The authors have provided a new experiment demonstrating that TRM improves classification performance by mitigating unknown inter-variable dependencies.
- Additional details have been included to clarify the experimental setup, particularly regarding hyperparameters for comparisons, allowing for better reproducibility and a clearer understanding of the methodology, including the handling of the time variable.

See the coment for further explanations.

**Justification Of The Preliminary Rating:**

The paper addresses a widespread and important problem in the medical context. It presents several strong points, such as the use of a well-established contrastive loss function, the incorporation of pre-trained models to enhance performance, and the application of masked attention mechanisms.
The paper is well-structured, making it easy to read and understand. However, as mentioned, some aspects of the methodology remain unclear—particularly the use of templated sentences despite the availability of structured clinical data. Additionally, certain methodological details are lacking, making it difficult to fully grasp and analyze the results.
I hope the discussion will help clarify these points.

**Questions To Address In The Rebuttal:**

- The first key question is whether template sentences add meaningful value beyond the clinical data itself.
- An ablation study would be valuable, for instance, using a simple MLP to encode clinical data and performing contrastive learning directly in this latent space.
- The authors should provide more details on the experimental impact of batch size in experiments and the results obtained.
- The authors should provide further explanations and data on the pre-training phase (losses, top k…) and the poor zero-shot performance that follows almost directly from it.
- The time variable handling in the model should be clarified.
- Description of evaluation methods and hyperparameter choices, particularly in the *Fine-tune Classification* (Section 3.2) and *Baselines and Ablations* (Section 3.3) should be improved.
- Finally, are the modalities truly unpaired in this case? The paper states that *"clinical variables are chronologically out of sync with the collected imaging in the NLST for up to four years in some cases."* However, many of these variables remain relatively stable over time—such as family history, chronic diseases, and, to some extent, smoking status. Additionally, variables like age, time to follow-up, and lung cancer labels can be directly inferred from the imaging acquisition date. Given this, the degree of misalignment and its actual impact on the analysis should be further clarified.

---

> ### Author Response · Authors · 2025-03-08
>
> **1.  My main concern with this work is the handling of clinical data as text.**
> While we agree with the intuition that tabular data should be used vectorized directly, our results suggest that contrastive self supervision has difficulty learning image-tabular representations where the inter-variable dependencies are unknown. In contrast, the TRM approach adds additional context to the clinical variables by allowing us to embed column names into a representation where the inter-variable dependencies are known via the semantic priors from open-ended web scale training. One of our contributions is demonstrating how this enables learning of multimodal representations in a lung screening population. We provide an additional ablation study (CLIP-tabular) where we performed CLIP on images and tabular data. After fine-tuning, this approach underperformed compared to the TRM-enabled CLIP model.
>
> **2. Clarification on evaluation methodology and implementation for baselines and ablations.**
>
> Additional detail is added under these sections.
>
> **3.  Regarding Section 3.1 (Zero-shot Classification): as acknowledged by the authors, this is not truly zero-shot, since the cancer label and follow-up time are included in the training process. Given this setup, the reported poor performance is particularly concerning. Once these variables are available, simple discriminative rules should suffice to determine the classification label at the 2-year mark. This raises questions about the effectiveness of the pre-training process. Are the validation loss functions behaving as expected? Additionally, top-k metrics are necessary to assess whether the training is properly aligning embeddings by bringing patient representations closer together.**
> As mentioned in the discussion, we expected poor zero-shot performance due to the limited concept-diversity of our data.  Our results suggest that the patient-level embeddings are not easily separable into the diagnostic classes by cosine similarity alone. Instead, a non-linear classifier, e.g. a finetuning stage, is required to fully leverage the utility of the learned representation. We did not aim to develop a foundation model with rich semantic understanding because this study was limited to images and 13 clinical variables.

---

> > ### Comment · Reviewer_YqoU · 2025-03-12
> >
> > - The authors indicate that the use of TRM rather than direct clinical data avoids the problem of unknown inter-variable dependencies. This argument is supported by a new experiment showing that the use of TRM improves classification performance.
> >
> > - As requested, the authors have added elements to clarify the experiments carried out, in particular the hyperparameters for the various comparisons.
> >
> > - The handling of the time variable in the TDViT has been described in the revised manuscript (through an embedding).
> >
> > - Regarding zero-shot classification, I acknowledge the authors' response, which is also explained in the manuscript. However, my question was more general and focused on the training process itself. Specifically, I am interested in whether the pre-training effectively brought the embeddings of the two modalities closer together for the same patient. Were modality retrieval performances (e.g., top-1, top-3 accuracy) on the validation set assessed, and did they indicate that the pre-training achieved its intended goal—namely, aligning the embeddings of different modalities?
> > Figure 3 does not explicitly show whether, for a given patient, the embedding of clinical variables and the embedding of the corresponding CT scan are actually close in the learned space.
> >
> > Since most of my questions have been addressed in the responses and the revised manuscript, I have decided to raise my score to 4 Weak Accept (see original review updated), while awaiting clarification on the final point I mentionned.
> >
> > Finally, while the authors have provided a primary answer, I believe that, beyond the scope of this paper, a deeper investigation would be valuable into the actual benefits of using template sentences that do not introduce new information, beyond a model architecture or a single experiment.

---

> > > ### Author Response · Authors · 2025-03-15
> > > **Response to Reviewer YqoU (round 2)**
> > >
> > > We greatly appreciate your additional round of feedback. We offer the following responses:
> > >
> > > **Regarding zero-shot classification, I acknowledge the authors' response, which is also explained in the manuscript. However, my question was more general and focused on the training process itself...**
> > >
> > > We see improvement and convergence of the validation loss over pre-training, but the final contrastive accuracy is quite low. The top-1 and top-10 accuracy of CLIP on the validation set at our selected checkpoint was 0.08 and 0.23 respectively. The top-1 and top-10 accuracy on the training set was 0.45 and 0.94. Another explanation for why the zero-shot classification was poor is that our transformer may be overfitting to the templates during pre-training, which are different than the label’s sentence template used at evaluation. We hope to address this in future work by co-optimizing the TRM, adding sentence template diversity, and training on a larger variety of lung-cancer related concepts.
> > >
> > > **Finally, while the authors have provided a primary answer, I believe that, beyond the scope of this paper, a deeper investigation would be valuable into the actual benefits of using template sentences that do not introduce new information, beyond a model architecture or a single experiment.**
> > >
> > > We agree with this sentiment. Sentence templates are one of many tabular serialization techniques that have been explored as large language models for tabular modeling has become more popular [1]. We experimented with JSON serialization (each row represented as a key/value array) with no success, but this could be a specific consequence of our choice of TRM.
> > >
> > > [1] X. Fang et al. Large Language Models(LLMs) on Tabular Data: Prediction, Generation, and Understanding -- A Survey, (2024). http://arxiv.org/abs/2402.17944

---

### Official Review · Reviewer_Sxcc · 2025-02-22

**Confidence:** 4
**Preliminary Rating:** 4

**Summary:**

This paper presents a novel approach called contrastive patient-level pretraining (CPLP) for fusing longitudinal chest CT images with clinical variables to predict lung cancer risk using a large public lung cancer screening dataset. The method leverages a time-distance transformer for imaging and a text representation model for clinical variables. The results show that CPLP, when fine-tuned, achieves better performance compared to conventional supervised approaches for lung cancer risk prediction.

**Strengths:**

1.The paper shows a novel application of contrastive learning to align unpaired modalities in real clinical settings.

2.This paper has done a thorough evaluation of two different clinical test sets: the entire test set (screening) and the subset of patients with a lung nodule detected (detected-nodules).

**Weaknesses:**

1.The paper lacks some detailed ablation studies on architectural choices and hyperparameters.

2.Limited discussion of potential biases in the training data. More details could be discussed in this paper.

**Detailed Comments:**

1. Can the authors quantify how missingness impacts performance (e.g., AUC per missing variable)? How does the model handle missing data in the clinical variables, beyond simply leaving them out of the paragraph?

2.The authors mentioned they used a two-sided Wilcoxon signed-rank test to do the statistical difference in terms of AUC among various approaches. However, the Wilcoxon test seems to be used for paired comparisons of two groups, not for comparing multiple models? Also, did the authors make some corrections during the statistical analysis?

**Justification Of The Preliminary Rating:**

The paper presents a novel approach to multimodal fusion in clinical settings on lung cancer risk prediction using a large public lung cancer screening dataset. The authors found that the CLIP representation significantly improves the prediction of lung cancer
risk in two types of clinical populations compared to conventional multimodal approaches. Nevertheless, there is room for improvement in several areas.

**Questions To Address In The Rebuttal:**

1.Section 2.1 states ResNet/TRM weights were frozen “to accommodate our desired batch size.” Was this decision purely computational, or did the authors empirically verify that fine-tuning would not improve performance? Can the authors share any ablation studies comparing frozen vs. trainable weights?

2. The authors identified 2401 as the “critical batch size” through ablation. Can the authors provide the exact AUC values and training dynamics observed for other batch sizes (e.g., 512 vs. 1024 vs. 2401)?

3. Which platform did the authors train/test the models? Did the authors use any libraries/frameworks? How many GPUs are used? Which operating system is used? The authors need to mention them in their paper.

---

> ### Author Response · Authors · 2025-03-08
> **Response to Reviewer Sxcc**
>
> 1. Can the authors quantify how missingness impacts performance (e.g., AUC per missing variable)? How does the model handle missing data in the clinical variables, beyond simply leaving them out of the paragraph?
> As a dataset curated from a research study, the NLST is fairly complete. We added an asterisk next to BMI and COPD, which were the two variables that some subjects were missing. For both variables, they were missing in less than 1% of subjects. Given this low degree of missingness, we did not analyze how the performance was impacted. For the multimodal-tabular model, we use multiple imputation to infer missing clinical variables under the assumption that they are missing at random.
>
> 2. The authors mentioned they used a two-sided Wilcoxon signed-rank test to do the statistical difference in terms of AUC among various approaches. However, the Wilcoxon test seems to be used for paired comparisons of two groups, not for comparing multiple models? Also, did the authors make some corrections during the statistical analysis?
> The reviewers are correct in that the Wilcoxon test compares difference between two groups. We conducted the statistical test between CLIP-finetune and each other method for a total of 6 total tests (CLIP-tabular was added to the result) in each test set. The result remained significant after Bonferroni correction of p=0.05.
>
> 3. Section 2.1 states ResNet/TRM weights were frozen “to accommodate our desired batch size.” Was this decision purely computational, or did the authors empirically verify that fine-tuning would not improve performance? Can the authors share any ablation studies comparing frozen vs. trainable weights?
> This decision was indeed made for computational reasons. Optimizing the ResNet/TRM parameters allocates around 16GB per example during training. Since having a sufficiently large batch size for CLIP is crucial, we would need to accumulating gradients across sub-batches to achieve our desired batch size, which was prohibitively time-intensive on our current hardware.
>
> 4. The authors identified 2401 as the “critical batch size” through ablation. Can the authors provide the exact AUC values and training dynamics observed for other batch sizes (e.g., 512 vs. 1024 vs. 2401)?
> The reason we did not provide a breakdown of fine-tune performance over CLIP batch size was because we observed model collapse during CLIP pretraining for batch sizes not sufficiently close to 2401. For instance, batch sizes of 1000, 5000, and 10,000 led to the loss converging close to a specific value early on with little change over a large number epochs (>3000). Based on previous studies, we suspect this is a pattern of model collapse, which occurs when the model converges on a trivial solution without reaching a meaningful representation.
>
> 5. Additional implementation details.
> Thank you for your feedback on the need for providing additional implementation details. We have added this to the Experiments and Results section.

---

> ### Comment · Area_Chair_nCzB · 2025-03-14
> **Please provide final rating based on the authors' rebuttal**
>
> Please provide final rating based on the authors' rebuttal. Thank you!

---

### Official Review · Reviewer_yQqs · 2025-02-26

**Confidence:** 4
**Preliminary Rating:** 3
**Recommendation:** Poster
**Final Rating:** 3

**Summary:**

This paper describes an approach where contrastive pre-training is applied to longitudinal and multimodal data.
The authors hypothesize that representations learned through contrastive patient-level pretraining would be more useful in lung cancer classification compared to representations learned through supervised learning. To investigate this, a CLIP representation is trained using clinical notes, and all CT scans, using NLST data.
The results show a bad performance for zero-shot performance, but after supervised fine-tuning, the results get a lot better, with a small margin over the baseline approaches.
The experiments were conducted in two cohorts: the entire test set (screening) and the subset of patients with a lung nodule detected (detected-nodules).
Finally, the authors conclude with the following claim "In data-limited settings, contrastive patient-level pretraining facilitates multimodal fusion, mitigating the overfitting challenges typically encountered with supervised approaches."

**Strengths:**

- Using contrastive learning for lung cancer classification from longitudinal multimodal data is novel.
- Relevant other work in this area is mentioned.
- Clear experimental setup.
- Publicly available code.

**Weaknesses:**

- The obtained AUCs are quite high for 2-year lung cancer risk estimation. The final approach reaches 0.895. Google AI reached an AUC of 0.873 using supervised CNNs in 2019, see Figure 4, Ardila et al. Nature Medicine, 2019. I find it difficult to believe that this approach beats a supervised CNN approahc trained with a lot of data. Can the authors comment on this? Different datasets?

- The train and test datasets are not clearly described. How did the authors create the datasets? Were they taken from previous publications, and are they constructed by the authors themselves? When 2-year lung cancer risk prediction is measured and all NLST CTs are used, does this mean that all lung cancers within 2 years after the last screening CT are included?

- Details about the datasets are missing. How many cancers were there in the test set, both for screening and for detected-nodule?

- I would like to see the claim that multimodal models more often overfit better explained. I do not understand why this would be the case, and I find the evidence in the paper for this. Can the authors add more references that support this claim?

- Unclear how the clinical notes using during CLIP training exactly looked like. Can the authors add a full example (Figure 1 does not contain the full list of clinical notes)?

**Detailed Comments:**

- What is a time distance transformer? How does it differ from a regular transformer?

- GitHub repo with code contains no license. This makes re-use of the code problematic. Please add a proper license, for example MIT license.

- A comparison to Google's CT foundation model which tried to same task would have been very interesting. Is this possible? The NLST embeddings have been released by the Google authors.

**Justification Of The Final Rating:**

I would like to thank the authors for the responses.

I am not sure I am fully convinced about the answers to my first point about the high AUC values. The mentioned second advantage of this approach is that Sybil is a strong imaging encoder because "trained on data outside of NLST" is I think simply not true. Sybil has been trained with NLST data only, see https://pubmed.ncbi.nlm.nih.gov/36634294/. Also, the authors are talking about the same test set for Sybil and Ardila et al., but even if the same test set patients were used, the evaluations are different: Sybil was evaluated for 6 year future lung cancer risk prediction, while Ardila et al. only looked at 1year and 2year lung cancer prediction, so I think the mentioned AUCs are not comparable.

I understand the time distance transformer and the text embeddings better now. However, compared to CLIP which had to learn the diversity of text, the provided text here comes from templated sentences, so is substantially different and less challenging.

All in all, I am sticking with my initial rating.

**Justification Of The Preliminary Rating:**

The methodology in this paper is novel, paper is written well, and the experiments are well described. However, there are too many details unclear about the data used for the experimens, and about the final AUC results to give a higher rating.

**Questions To Address In The Rebuttal:**

Please address my comments about the dataset descriptions, the relatively high AUCs in comparison to the work by Google AI, and the claims about multimodals more often overfitting.

**Special Issue:**

No

---

> ### Author Response · Authors · 2025-03-08
> **Response to Reviewer yQqs**
>
> 1. AUCs are high compared to Ardila et al.
> Ardila et al. developed a CNN that operated on up to two chest CTs and was trained on the NLST. In contrast, our classification model is pretrained using CLIP and operates on up to three chest CTs and clinical variables. We use the same training set, thought there may be slight differences in inclusion criteria, and evaluate on the same test set. We cite three advantages with our approach that may explain the better performance. The first advantage is CPLP, which is the main contribution of this paper. Second, we leverage Sybil, an open-sourced ResNet, as our imaging encoder which performs on par with Ardila et al. on the same test set (0.86 vs 0.87 respectively), but has the advantage of being trained on data outside of the NLST. Third, our approach benefits from the added predictive power of clinical variables. Consider that the Mayo model[1], using clinical variables only, reaches a relatively competitive classification at 0.80 AUC [2] on this test set. To our knowledge, this study is the first multimodal approach that outperforms Sybil and Ardila et al. in the NLST. Outside of the NLST, several previous studies have leveraged multimodal prediction to improve lung cancer risk estimation over imaging-only models [3–6] although having these models generalize outside of their development distribution is an open challenge.
>
> 2. Additional detail on training a test splits
> We have added detail on how train and test splits were created.
>
> 3. Additional details on cancer prevelance.
> We updated Table 1 to include the number of patients who developed cancers after 2 years.
>
> 4. Multimodal models overfitting?
> We have noticed that in practice, multimodal models have a greater tendency to overfit (high train accuracy but lower validation and test accuracy) compared to their single modality counterparts if all else is constant due to two main reasons. First, training multimodal models involves fitting an increased number of input features using the same amount of data. When investigating medical datasets of smaller size, the classic p>n problem leads to overfitting on the training distribution, at least compared to simpler models. Second, (Wang et al, 2020) found that modalities tend to converge at different rates during optimization, such that the gradients for one modality are overfit while the gradients for another have yet to converge. We’ve added additional citations to the introduction an discussion that cover this idea of multimodal overfitting. We limit these claims to the domain of medical classification and refrain from drawing the same conclusions for other domains.
>
> 5. Unclear how the clinical notes using during CLIP training exactly looked like.
> 	To clarify, we did not use clinical notes in this study. The only non-imaging data were the 13 clinical variables listed in Table 1.
>
> 6. What is a time distance transformer? How does it differ from a regular transformer?
> 	The time-distance transformer refers to simple extension of transformers that add time encodings to the input embeddings, providing longitudinal context. We rephrased the Image Encoder section to clarify how  time encodings were used to specify the time interval between scans.
>
> 7.  Please add a proper license, for example MIT license.
> Thank you for brining this to our attention. A license has been added.
>
> 8.  A comparison to Google's CT foundation model which tried to same task would have been very interesting. Is this possible? The NLST embeddings have been released by the Google authors.
> Thank you for bringing Google’s CT Foundation to our attention. We agree that evaluating these embeddings by finetuning for lung cancer risk prediction would be an interesting approach. As we applied for an API key and look forward to the comparison in future work.
>
> [1]	S.J. Swensen et al. The Probability of Malignancy in Solitary Pulmonary Nodules: Application to Small Radiologically Indeterminate Nodules, Arch Intern Med 157 (1997) 849–855.
> [2]	T.Z. Li et al. Performance of Lung Cancer Prediction Models for Screening-detected, Incidental, and Biopsied Pulmonary Nodules, Radiol Artif Intell (2025).
> [3]	R. Gao, et al. Deep multi-path network integrating incomplete biomarker and chest CT data for evaluating lung cancer risk, Https://Doi.Org/10.1117/12.2580730 11596 (2021) 387–393.
> [4]	R. Gao, et al. Lung Cancer Risk Estimation with Incomplete Data: A Joint Missing Imputation Perspective, Lecture Notes in Computer Science (Including Subseries Lecture Notes in Artificial Intelligence and Lecture Notes in Bioinformatics) 12905 LNCS (2021) 647–656.
> [5]	R. Gao, et al. Cancer risk estimation combining lung screening ct with clinical data elements, Radiol Artif Intell 3 (2021).
> [6]	T.Z. Li, et al. Longitudinal Multimodal Transformer Integrating Imaging and Latent Clinical Signatures from Routine EHRs for Pulmonary Nodule Classification, Lecture Notes in Computer Science

---

### Author Rebuttal · Authors · 2025-03-08

**Rebuttal:**

We appreciate the feedback from the reviewers and address each in the comments section below along with highlighted changes in green to the manuscript.

**Supporting Material:**

/attachment/3a87a0e7d134da8faba515ab4089dd2a2772c639.pdf

---

### Meta-Review · Area_Chair_nCzB · 2025-03-19

**Recommendation:** Accept (Oral)
**Confidence:** 5

**Metareview:**

This manuscript pursues a contrastive language-image pretraining-based approach to classifying 2-year lung cancer risk, adding a time-distanced transformer to encode longitudinal information from sequential screening exams. The work has a number of strengths, including incorporating imaging, text, and temporal information into the model and a thorough evaluation across multiple test sets. Reviewer comments were generally positive, and the authors were seen to be responsive. One reviewer raised the point about performance compared to Ardila et al. The author's response seems reasonable, and it should be noted that their approach is also trained on a larger  NLST cohort, 61,000 CTs from 22,571 patients (versus 42,290 CTs in Ardila et al). That said, it is difficult to ensure whether the difference in performance will generalize, especially since Ardila et al. did not open-source their model. Note that the manuscript references (Figure 2, CLIP-zero-shot) probably should refer to Table 2, not Figure 2.